# Integrating Molecular Alterations with Immunophenotype and Clinical Characteristics in Myelodysplastic Syndromes: A Single-Center Study

**DOI:** 10.3390/ijms26157382

**Published:** 2025-07-30

**Authors:** Maciej Majcherek, Krzysztof Przeorski, Aleksandra Mroczkowska-Bękarciak, Natalia Nogaj, Donata Szymczak, Anna Kopszak, Krzysztof Kujawa, Paula Jabłonowska-Babij, Maciej Tomasiewicz, Agnieszka Szeremet, Tomasz Wróbel, Anna Czyż

**Affiliations:** 1Clinic of Hematology, Cellular Therapies and Internal Medicine, Wroclaw Medical University, 50-367 Wrocław, Poland; krzysztof.przeorski@usk.wroc.pl (K.P.); aleksandra.mroczkowska-bekarciak@student.umw.edu.pl (A.M.-B.); donata.szymczak@umw.edu.pl (D.S.); paula.jablonowska@usk.wroc.pl (P.J.-B.); maciej.tomasiewicz@student.umw.edu.pl (M.T.); agnieszka.szeremet@umw.edu.pl (A.S.); tomasz.wrobel@umw.edu.pl (T.W.); a.czyz@umw.edu.pl (A.C.); 2Laboratory of Flow Cytometry and Cytomorphology, Department of Hematology and Bone Marrow Transplantation, University Hospital in Wroclaw, 50-556 Wroclaw, Poland; natalia.nogaj@usk.wroc.pl; 3Statistical Analysis Centre, Wroclaw Medical University, 50-367 Wroclaw, Poland; anna.kopszak@umw.edu.pl (A.K.); krzysztof.kujawa@umw.edu.pl (K.K.)

**Keywords:** NGS, MDS, Ogata score, Flow Cytometry, MFC

## Abstract

Continuous development of molecular and immunophenotypic techniques enables more precise diagnoses and more accurate assessment of prognosis in myelodysplastic syndromes (MDS). However, the relationship between genetic alterations and immunophenotype remains very poorly understood. The analysis included 30 patients diagnosed at a tertiary center who were eligible for azacitidine treatment. Next-generation sequencing (NGS) was performed at the start of the study to assess the mutation status of 40 genes associated with MDS pathogenesis. In addition, multiparametric flow cytometry (MFC) was performed to assess the ELN score (Ogata score) and, additionally, to detect an abnormal CD11b/HLA-DR and CD11b/CD13 expression pattern. In the studied patient population, higher ELN score results were found in patients with mutations in epigenetic modifiers and pathogenic mutations of the tumor suppressor genes. Signal pathway mutations were associated with lower platelet counts at diagnosis. The results of this study indicate a correlation between molecular abnormalities and deviations in cell immunophenotype. Investigating this correlation may, in the future, allow the development of new scales that allow a more sensitive and specific diagnosis of MDS and a more precise prediction of its course.

## 1. Introduction

Despite the continuous development of diagnostic tools such as cytogenetic studies, molecular, histopathology, and flow cytometry, the diagnosis and prognosis of the MDS still remains difficult and inconclusive. One of the most important problems clinicians are currently facing is distinguishing clonal disorders of hematopoiesis from secondary causes of cytopenias (B12 deficiency, immunological). Another problem is also the occurrence of new phenomena such as clonal hematopoiesis of undetermined potential (CHIP) and clonal cytopenia of undetermined significance (CCUS) [1,2].

Advances in the diagnosis of MDS are brought by the increasingly widely used next-generation sequencing (NGS) techniques. This is reflected in the latest WHO and ICC 2022 classifications, which include several new categories of MDS whose criteria are based on NGS results [3].

Considerable progress was also made with the development of new increasingly useful testing algorithms and scores using flow cytometry to diagnose MDS [4,5].

In addition to their diagnostic importance, both NGS and FCM tests have significant value in assessing the risk of disease progression [6].

The key to success in the diagnosis of MDS, however, seems to be the joint use of different complementary techniques that can lead to a unified system that binds the various methods useful in the diagnosis of MDS [7,8].

In the following paper, we present our observations on combining flow cytometry with molecular biology techniques with treatment-naive MDS patients. We compare the molecular abnormalities associated with the pathogenesis of MDS with the phenotype of marrow cells.

## 2. Results

The study included 30 patients diagnosed with MDS, the majority of whom had intermediate, high, or very high risk and were candidates for treatment with azacitidine.

All the patients were diagnosed and treated at a single referral center. Most patients (87%) received at least one cycle of azacitidine treatment. Detailed characteristics of the group are provided in Table 1. Cytogenetic results were available in 28 (93%) patients. CMML patients were not classified according to the R-IPSS criteria. The collective results of the NGS test and the results obtained in the ELN 2012 score and the extended ELN 2012 score [9] are presented in Table 2. The remaining results can be found in the Appendix A. Figure 1 presents the summary results of the parameters obtained during the evaluation according to the ELN2012 score. Detailed information on mutations and their relationship with ELN 2012 score is given in in Appendix A and the results of individual parameters within the ELN 2012 score in Appendix A. Appendix A contains the results of correlation of individual parameters in the entire study group.

First, the relationship between the presence of mutations detected by NGS and the ELN 2012 score was sought. The presence of tumor suppressor gene mutations was found in a total of six patients, including pathogenic TP53 mutations in five of them. In our group, all but one patient with a TP53 mutation had a complex karyotype. Higher ELN 2012 scores were found (*p* = 0.045) among patients with TP53 mutation. The median ELN 2012 score for this group was 3 (range 1–4), while among the rest of the patients it was 2 (range 2–3). In the case of the remaining molecular groups (signal pathway, transcriptional factor, mRNA splicing, cohesin complex mutation), no significant correlations with the ELN 2012 score were revealed. Of the 30 patients studied, 20 had at least one pathogenic epigenetic regulation mutation. The patients with epigenetic regulation mutations were presented with higher ELN 2012 scores compared to patients who did not show these mutations [median 3 (range 2–4) vs. median 2 (range 1–4); *p* = 0.044].

In the next step, correlations between the presence of mutations and complete blood count values were investigated. Patients with signal pathway mutations group (*n* = 7) had a lower platelet count compared to the other patients (median 50 × 10^3^/uL, range 7–129 vs. median 73 × 1000/uL, range 21–241; *p* = 0.045). Additionally, the patients with tumor suppressor mutations also presented with lower platelet count compared to the remaining individuals (median 48 × 10^3^/uL, range 7–79 vs. median 69 × 10^3^/uL, range 21–24; *p* = 0.034].

Moreover, patients with cohesin complex mutations (*N = 3*) had lower leukocyte counts at diagnosis (median 2.08, range 1.57–3.54) compared to the remaining individuals (median 3.31, range 1.2–24.5) (*p* = 0.026).

Finally, correlations between the values of the ELN 2012 score and R-IPSS and M-IPSS, as well as the relationship between the R-IPSS value and mutation status were considered. No statistically significant association was found between prognostic scales and ELN 2012 score.

## 3. Discussion

Despite the availability of an increasing number of very precise tools, MDS diagnostics still cause a lot of problems. Currently, there is no single tool that allows for an unequivocal diagnosis of MDS. Both FCM and NGS testing have their strengths and weaknesses. FCM is quickly available and much cheaper, but less sensitive and specific [9,10]. On the one hand, the detection of pathogenic mutations in patients with cytopenia indicates a suspicion of MDS, but phenomena such as CHIP, ICUS, and CCUS make it difficult to interpret. Some publications show that combining multiple techniques can improve the quality of diagnostics and prognosis of the course of the disease [11]. There is a growing amount of evidence that MDS-associated mutations detected by NGS can affect the phenotype of neoplastic cells in both MDS and AML [12]. In our study, we explored a relationship between the occurrence of single mutations detected by NGS and the immunophenotype of bone marrow cells. We were inspired by the data published by Wiess et al., who showed the influence of the presence of SRSF2 mutations on the expression pattern of CD16/CD11b on granulocytes [7]. In our work, we have proven that the presence of pathogenic mutations of the TP53 gene are associated with a higher ELN 2012 score. Such a phenomenon has not been described so far. Of note, both the presence of TP53 mutations and a higher ELN 2012 score are associated with the risk of a worse course of the disease, which can partially explain the association between these phenomena [6,13].

Due to the considerable number of different mutations occurring in MDS, in some earlier papers they are joined into groups based on similar mechanisms of influence on cell physiology and development of MDS [13]. We have shown that mutations associated with epigenetic modification (DNMT3A, TET2, and IDH1/2) are associated with a higher ELN score. The mutations from the above group were previously shown to be associated with the abnormal immunophenotype. Moreover, Guarneri et al. confirmed that patients with an ELN score > 2 were more likely to have SRSF2 and TET2 mutations [14].

We also found some associations between complete blood count and some mutations. According to the results of our study, lower platelet counts correlate with the presence of signal pathway and tumor suppressor gene mutations. Lower platelet counts have been previously reported by other authors in patients with TP53-mutated MDS [15]. There have also been reports that lower platelet count at diagnosis is associated with a worse prognosis, similarly to the previously mentioned TP53 mutation [16]. We also showed that the presence of the cohesin complex mutation was associated with lower leukocyte count. However, due to the very low number of patients showing this mutation, we treat this result with caution. On the other hand, the presence of cohesin complex mutations is associated with a higher incidence of more advanced MDS subtypes and a faster progression to AML, which may indicate a relationship between the number of leukocytes and cohesin complex mutations [17].

A surprising finding for us was the fact that we did not find a correlation between the R-IPSS or M-IPSS result and the ELN 2012 score. Many of the previously presented publications and our own experience have indicated that higher ELN 2012 scores are associated with higher R-IPSS values [9,18,19]. Moreover, Oelschlaegel et al. showed that parametric scores can predict survival time even better than R-IPSS [6]. One reason no such correlations were shown in our current study is that we qualified mainly intermediate- and high-risk patients according to the R-IPSS who required azacitidine treatment. In addition, the relatively limited number of patients in our study may make it difficult to statistically confirm such a correlation.

It should be noted that, as in other cited works, our study included a very small patient group, which may significantly impact the statistical power of the results. Therefore, the findings should be interpreted as indicative, providing direction for future research rather than definitive conclusions. Additionally, it would be advisable to incorporate an analysis of clinical parameters such as age, gender, transfusion dependence, and time to disease progression. Given the occurrence of phenomena such as CHIP and CCUS, it would also be valuable to compare our results with those from a validation cohort, for example, healthy individuals or patients diagnosed with CHIP/CCUS. However, the invasiveness of bone marrow biopsy and the high cost of diagnostics present considerable challenges to collecting such a validation group.

Due to the development of both FCM and NGS techniques, MDS diagnosis algorithms seem to be increasingly complicated and difficult to implement in everyday practice. Therefore, the use of Artificial Intelligence (AI) and Machine Learning (ML) techniques in analyzing data obtained during MDS diagnostics is gaining importance [20]. There have already been reports of excellent sensitivity and specificity of MDS diagnoses based on scales developed using AI. Further development of NGS, MFC, and AI-based data processing techniques may lead to increasingly better diagnostic and prognostic results for MDS in the future [21,22].

In conclusion, it should be emphasized that the presence of specific mutations affects both the immunophenotype of cells and the clinical picture of MDS. Despite the inherent limitations of a single-center study, we believe that our findings provide valuable insights and have the potential to contribute to the development of improved diagnostic and prognostic strategies for myelodysplastic syndromes. Further analysis of NGS and FCM data may lead to the improvement of existing or the creation of new tools for diagnostics and prediction of the clinical course of the disease. However, drawing conclusions from such a large amount of data provided by NGS and FCM seems to require the use of modern methods of data analysis, such as ML and AI.

## 4. Materials and Methods

### 4.1. Patients and Medical Data

This single-center prospective study included previously untreated patients diagnosed with myelodysplastic syndrome according to WHO 2016 criteria [23] who were eligible for azacitidine treatment between 2021 and 2024 at our center. Patients were qualified for treatment according to European Society for Medical Oncology (ESMO) 2021 recommendations and gave informed consent to participate in the study. The study was reviewed by the local ethical board at Wroclaw Medical University [24]. Relevant clinical data on diagnosis, age, sex, transfusion dependency, hemoglobin, leukocyte and platelet levels, bone marrow myeloblast count, survival, R-IPSS [25], and M-IPSS [26] were also collected.

### 4.2. Diagnostic Procedures

The material for the study was bone marrow. The first collection by aspiration was performed when MDS was suspected. During this collection, a smear, FCM test, classical cytogenetics, and FISH were performed, and the material was stored for NGS. A bone marrow smear was performed according to the standard procedure.

### 4.3. Flow Cytometry

Biopsy samples were transported and processed within an hour of collection. For FACS analysis, 4.9 mL of bone marrow was collected into EDTA tubes (Sarstedt, Nümbrecht, Germany)—the first portion to minimize hemodilution. The study was conducted using two panels consisting of 8 antibodies, using the stain–lyse–wash method, analyzed on a FACS Canto II BD flow cytometer (Becton Dickinson and Company, San Jose, CA, USA). Lysing Solution BD was used for lysis (diluted 10 times). The current work uses the results from the first test tube. The following mouse anti-human monoclonal antibodies, purchased from BD and BC (Beckman Coulter Brea, CA, USA), were used for analysis: CD15, CD11b, CD34, CD117, CD10, HLA-DR, CD13, CD45. Cells were surface stained with the following fluorescence conjugated mouse anti-human monoclonal antibodies in one eight-color tube: anti-CD15 FITC clone MMA, anti-CD11b PE clone D12, anti-CD34 PerCP-Cy5.5 clone 8G12, anti-CD117 PC-7 clone 104D2D1, anti-CD10 APC clone HI10a, anti-HLA-DR APC-H7 clone L243, anti-CD13 BV421clone WM15, anti-CD45 V500C clone 2D1. The evaluation of nucleated cells was carried out on an 8-color FACS Canto II flow cytometer (BD). In each test tube, we collected as many cells as possible. On average, there were 500,000 events. The data were analyzed using BD FACSDiva software v8.0—the gating strategy is shown in Figure 2.

The steps for ELN score assessment are illustrated in Figure 2A–H [27]. Immunophenotypic disorders of leukocytes in the bone marrow were detected by eight-color flow cytometry to assess the ELN score value extended with immunophenotypic disorders significantly we found more common in MDS, i.e., abnormal maturation patterns of expression of antigens CD11b vs DR and CD11b vs CD13 (shown in Figure 2G–J). The exact method is described in our published paper by our team [9].

### 4.4. DNA Isolation DNA

Isolation was performed by the column method using the Quick Blood DNA Purification Kit (Eurx). The purpose of this kit is to quickly isolate extremely pure genomic DNA. Proteinase K and specialized lysis solution were used to lyse the samples. Cellular proteins are broken down by proteinase K, which also removes DNA from all bound proteins, including nucleases. By adding ethanol to the lysate, the right conditions are generated for DNA to bond to the GeneMATRIX resin. DNA attaches to the spin-column’s silica membrane during the brief centrifugation process, but impurities flow through. Two wash procedures effectively eliminate any traces of impurities that may still be present on the resin. A low-salinity buffer was used to elute high-quality cellular DNA.

### 4.5. Next Generation Sequencing

NGS was performed on the Illumina MiniSeq instrument using the AmpliSeq for Illumina Myeloid Panel, which allows for the detection of molecular changes in 40 genes, 17 full genes (*ASXL1*, *BCOR*, *CALR*, *CEBPA*, *ETV6*, *EZH2*, *IKZF1*, *NF1*, *PHF6*, *PRPF8*, *RB1*, *RUNX1*, *SH2B3*, *STAG2*, *TET2*, *TP53*, *ZRSR2*), and 23 hotspot genes (*ABL1*, *BRAF*, *CBL*, *CSF3R*, *DNMT3A*, *FLT3*, *GATA2*, *HRAS*, *IDH1*, *IDH2*, *JAK2*, *KIT*, *KRAS*, *MPL*, *MYD88*, *NPM1*, *NRAS*, *PTPN11*, *SETBP1*, *SF3B1*, *SRSF2*, *U2AF1*, *WT1*). Library preparation and sequencing were performed in accordance with the manufacturer’s instructions. The purpose of this specific panel was to investigate somatic variations associated with hematological malignancies. Changes in the insertion, deletion, and substitution of several nucleotides can be detected with this panel. The Illumina Sequence Analysis Viewer tool was used to display sequencing metrics. The FastQC version v0.12.1 software from the Illumina BaseSpace^TM^ Sequence Hub was used to evaluate the quality of the raw NGS data. The sequencing data was examined using the Illumina BaseSpace^TM^ Sequence Hub, variant calling programs (Variant Interpreter), and sequence alignment (DNA Dragen Amplicon). The parameters for mutation calling were a minimum coverage of 500 called regions and a minimum variable allele frequency (VAF) of 5%. Aligned read (bam) and (.bam.bai) files for each sample were manually examined using the Integrative Genomics Viewer (IGV) program (Broad Institute, Cambridge, USA) to verify the existence of the filtered-in and prioritized variations. The genomic databases VarSome Clinical, COSMIC, Genoox-Franklin, dbSNP, ClinVar, and gnomAD were used to identify variants. Variants were classified as either pathogenic, likely pathogenic, or variant of uncertain significance (VUS).

Due to the rare occurrence of single mutations, we grouped them for further analysis according to their role in the pathogenesis of MDS. The following groups described earlier were distinguished [28,29]:-signal pathway mutations (KIT, NRAS, KRAS, JAK2, CSF3R, MPL)-mutations in epigenetic modifiers (TET2, DNMT3A, ASXL1, IDH1, IDH2)-transcription factor mutations (NPM1, RUNX1, GATA2)-tumor suppressor mutations (TP53, PHF6, NF1)-mutations in the cohesion complex (STAG2, IKZF1)-mRNA splicing factor mutations (U2AF1, SRSF2, ZRSR2, SF3B1, PRPF8)

### 4.6. Statistical Analysis

The statistical analysis consisted of examining the differences in the values of continuous variables, in groups based on the binary grouping variables and checking the correlations between continuous variables.

For the statistical analysis, the number of mutations in each group highlighted above was summed and, in addition, in each group separately, the number of mutations considered pathogenic or probably pathogenic. The resulting score was correlated with the FCM test results (ELN score and ELN extended with CD13/11b+ CD11b/HLA-DR).

For the comparisons between two groups, we first checked the data distribution of each continuous variable in these groups. When the distribution was not normal or the number of observations per group was too small to determine normality (<10 observations), we used the permutational Student’s t-test or the Mann–Whitney U test instead of the standard Student’s t test. In some cases, the more powerful permutational Student’s t-test indicates that the differences are significant, but the Mann–Whitney test does not. These results should be interpreted as marginally statistically significant, which means that more studies are needed to fully assess the statistical significance of the measured effect.

The correlations between continuous variables were calculated using Kendall’s tau b coefficient due to the non-normal distribution of the variables and the presence of ties in the data.

The hypotheses were considered separately, and the results were not concluded jointly, so the multiple comparisons correction was not applied.

The local bioethics committee approved the study.

## Figures and Tables

**Figure 1 ijms-26-07382-f001:**
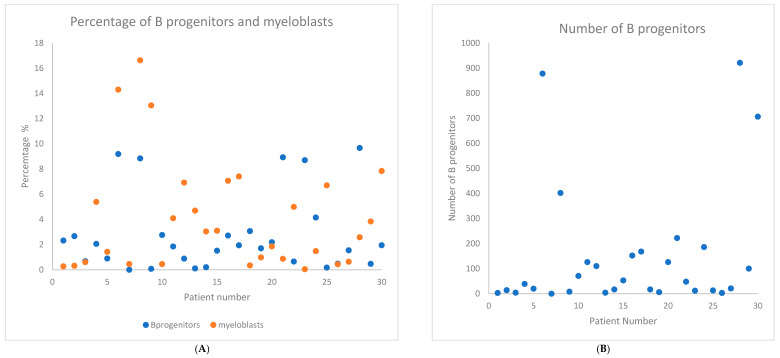
Values of individual parameters in the ELN 2012 score in each patient in the study group (axis X). (**A**) Percentage of B progenitors and myeloblasts. (**B**) Number of B lymphocytes. (**C**) Number of myeloblasts. (**D**) CD45 MFI on lymphocytes and myeloblasts. (**E**) Side Scatter Count (SSC) of lymphocytes and granulocytes.

**Figure 2 ijms-26-07382-f002:**
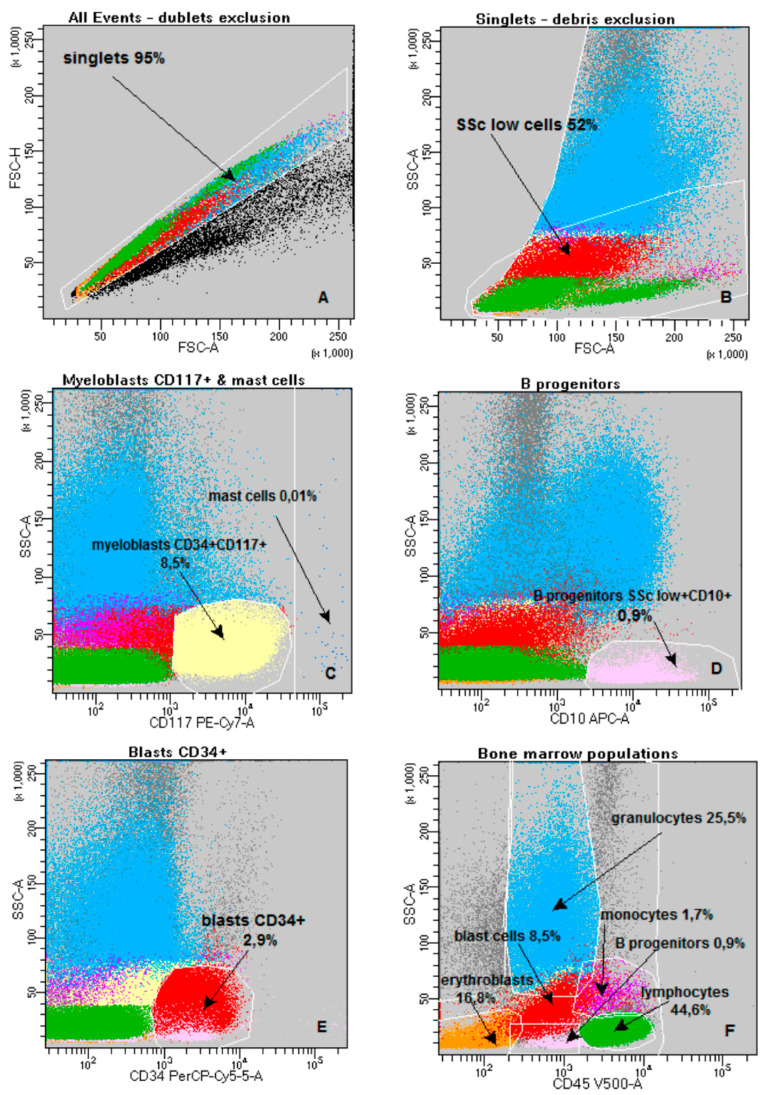
Eight-color flow cytometric analysis. (**A**–**G**,**I**) bone marrow of a patient with MDS—Ogata score 3 points plus an abnormal neutrophil maturation pathway. (**H**,**J**) Bone marrow of a patient with thrombocytopenia of other origin than MDS—Ogata score 0 points, comparison of neutrophil maturation with a patient with MDS. The gating strategy: (**A**)—discrimination of doublets (FSC-A vs. FSC-H)- singlets constitute 95% of all bone marrow cells; (**B**)—discrimination of cells debris and gating cells SSC low (FSC-A vs. SSC-A)—SSC low cells constitute 52% of bone marrow cells, after excluding debris and doublets; (**C**)—gating myeloblasts CD117+ and mast cells for quality control; (**D**)—gating B-cell progenitors (CD10 vs. SSC-A)—percentage among all bone marrow cells; (**E**)—gating blast cells CD34+ (CD34 vs. SSC-A)—percentage among all bone marrow cells CD45+; (**F**)—gating of the bone marrow populations with particular emphasis on CD45 positive cells (CD45 vs. SSC-A): blue—neutrophils, red—blast cells, purple—monocytes, pink—B progenitors, green—lymphocytes, orange—erythroblasts—percentages of the presented populations are described among all bone marrow cells; (**G**)—neutrophils with abnormal maturation pathway (CD11b vs. CD13); (**H**)—neutrophils with normal maturation pathway (CD11b vs. CD13); (**I**)—neutrophils with abnormal maturation pathway (CD11b vs HLA-DR); (**K**)—comparison of neutrophil percentages in specific gates marked on (**G**,**H**); (**L**)—comparison of neutrophil percentages in specific gates marked on (**I**,**J**), #Events—Number of events.

**Table 1 ijms-26-07382-t001:** Detailed characteristics of the group.

Baseline Characteristics	*N* (%) or Median (Range)
**Age in years**	68 (52–84)
Patient gender	
Female	13 (43%)
Male	17 (57%)
Diagnosis	
MDS	27 (90%)
CMML	3 (10%)
MDS WHO 2016 classification (*N = 27*)	
MDS-LB	4 (13%)
MDS-EB1	5 (17%)
MDS-EB2	18 (60%)
IPSS-R risk stratification (*N = 27*)	
Very low	2 (7%)
Low	3 (11%)
Intermediate	7 (26%)
High	7 (26%)
Very high	8 (30%)
Cytogenetic (*N = 28*)	
Normal karyotype	16 (57%)
Complex karyotype	5 (18%)
Del -7	3 (11%)
Del 9q-	1 (4%)
Del 11q-	1 (4%)
Del 5q- and del -13	1 (4%)
Trisomy +8	1 (4%)
FCM and treatment results	
Positive ELN score (>1) at diagnosis	30 (100%)

MDS—myelodysplastic syndrome; CMML—chronic myelomonocytic leukemia MDS-LD—Myelodysplastic syndrome with low blasts; MDS-IB—myelodysplastic syndrome with increased blasts; IPSS-R—Revised International Prognostic Scoring System; NGS—next-generation sequencing; VUS—variant of unknown significance; FCM—flow cytometry; del—deletion; ELN—European Leukemia Net.

**Table 2 ijms-26-07382-t002:** The results of the NGS and the results of the ELN 2012 score and the extended ELN 2012 score.

Baseline Characteristics	*N (%)* or Median (Range)
**NGS mutation status**	
Patients with any detectable mutation on NGS	27 (90%)
Patients with only VUS mutation on NGS	1 (3%)
Median number of mutations (range)	2.5 (0–6)
Median number of pathogenic mutations (range)	2 (0–4)
Number of patients with any signal pathway mutations (KIT, KRAS, JAK2, NRAS, CSF3R, MPL)	8 (27%)
Number of patients with pathogenic signal pathway mutations (KIT, KRAS, JAK2, NRAS, CSF3R, MPL)	7 (23%)
Number of patients with any epigenetic regulation mutations (TET2, DNMT3A, ASXL1, IDH1/2)	20 (67%)
Number of patients with pathogenic epigenetic regulation mutations (TET2, DNMT3A, ASXL1, IDH1/2)	19 (63%)
Number of patients with any mRNA splicing mutations (U2AF1, SRSF2, ZRSR2, SF3B1, PRPF8)	12 (40%)
Number of patients with pathogenic mRNA splicing mutations (U2AF1, SRSF2, ZRSR2, SF3B1, PRPF8)	6 (20%)
Number of patients with any transcriptional factor mutations (NPM1, RUNX1, GATA2)	9 (30%)
Number of patients with pathogenic transcriptional factor mutations (NPM1, RUNX1, GATA2)	8 (27%)
Number of patients with any tumor suppressor mutations (TP53, PHF6, NF1)	6 (20%)
Number of patients with pathogenic tumor suppressor mutations (TP53, PHF6, NF1)	5 (17%)
Number of patients with any cohesin complex mutations (STAG2, IKZF1)	7 (23%)
Number of patients with pathogenic cohesin complex mutations (STAG2, IKZF1)	4 (13%)
Number of patients with any other mutations (CBL)	1 (3%)
Number of patients with other pathogenic mutations (CBL)	1 (3%)
**FCM and treatment results**	
Positive ELN score at diagnosis (>1)	29 (97%)
ELN score at diagnosis.	
ELN score 1 at diagnosis	1(3%)
ELN score 2 at diagnosis	11(37%)
ELN score 3 at diagnosis	13 (43%)
ELN score 4 at diagnosis	5 (17%)
**Extended ELN score at diagnosis**. - 2	2 (7%)
- 3	7 (23%)
- 4	16 (53%)
- 5	3 (10%)
- 6	2 (7%)

NGS—next-generation sequencing; VUS—variant of unknown significance; FCM—flow cytometry; ELN—European Leukemia Net.

## Data Availability

Data available on request due to privacy/ethical restrictions.

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
