# Peer review of "Integrating Molecular Alterations with Immunophenotype and Clinical Characteristics in Myelodysplastic Syndromes: A Single-Center Study"

_ijms, 2025, doi:10.3390/ijms26157382_

Round 1

Reviewer 1 Report (New Reviewer)

Comments and Suggestions for Authors

The relationship between genetic mutations and immune phenotypes in myelodysplastic syndromes (MDS) has not yet been fully elucidated. In this study, we found the relationship between genetic mutations and immune phenotypes as shown in â‘  to â‘¢. â‘  Patients with epigenetic modifier mutations and pathogenic mutations in tumor suppressor genes were associated with high ELN scores in the immune phenotype. â‘¡ Mutations in signaling pathways were associated with low platelet counts. â‘¢ Patients with related cohesin complex mutations were associated with low white blood cell counts. â‘£ The presence of cohesin complex mutations is significant because it was revealed in this study to be associated with an increased incidence of more advanced MDS subtypes and a more rapid progression to AML. However, the following three points would be added to the study to make it even better. (1) The data that supports â‘  to â‘£ should be clearly shown. (2) What are the possible reasons for â‘  to â‘£? (3) Please provide any data on correlation between epigenetic modifier mutations, tumor suppressor genes, signal pathway mutations, cohesin complex mutations, transcription factor mutations, adhesion complex mutations, mRNA splicing factor mutations, chromosomal abnormalities, ELN score, CD11b/DR and CD11b/CD13 phenotypes, decreased platelet counts, decreased white blood cell counts, etc. and prognosis (e.g., survival time).

Author Response

The relationship between genetic mutations and immune phenotypes in myelodysplastic syndromes (MDS) has not yet been fully elucidated. In this study, we found the relationship between genetic mutations and immune phenotypes as shown in â‘  to â‘¢. â‘  Patients with epigenetic modifier mutations and pathogenic mutations in tumor suppressor genes were associated with high ELN scores in the immune phenotype. â‘¡ Mutations in signaling pathways were associated with low platelet counts. â‘¢ Patients with related cohesin complex mutations were associated with low white blood cell counts. â‘£ The presence of cohesin complex mutations is significant because it was revealed in this study to be associated with an increased incidence of more advanced MDS subtypes and a more rapid progression to AML. However, the following three points would be added to the study to make it even better. (1) The data that supports â‘  to â‘£ should be clearly shown. (2) What are the possible reasons for â‘  to â‘£? (3) Please provide any data on correlation between epigenetic modifier mutations, tumor suppressor genes, signal pathway mutations, cohesin complex mutations, transcription factor mutations, adhesion complex mutations, mRNA splicing factor mutations, chromosomal abnormalities, ELN score, CD11b/DR and CD11b/CD13 phenotypes, decreased platelet counts, decreased white blood cell counts, etc. and prognosis (e.g., survival time).

Dear reviewer, thank you very much for your valuable comments, we fully agree that the above proposals would increase the value of the work, but many of these proposals are not feasible for distinct reasons.

Referring to the proposals:

Reviewer: (1) The data that supports â‘  to â‘£ should be clearly shown.

Response 1. All data used to conduct the statistical analysis that we used in our publication is available in the supplement in tables 1-3. Of note, Supplementary Table 3 contains the results of correlation of individual parameters in the entire study group.

Reviewer: (2) What are the possible reasons for â‘  to â‘£?

Response: 2.  The possible reason for the association between the severity of immunophenotypic abnormalities and epigenetic mutations, as well as TP53 mutations in MDS, cannot be directly explained with the current level of knowledge. However, as we emphasized in the discussion, mutations in the epigenetic group have previously been shown to be associated with abnormal immunophenotypes. Additionally, both a high ELN risk score and TP53 mutations are linked to poor prognosis in MDS. This is certainly an interesting area for further research into the biology of MDS. Since our study is based solely on clinical data, we are not able to explain this phenomenon. Unfortunately, neither our data nor previous publications allow for even an approximate answer to this question.

Reviewer: (3) Please provide any data on correlation between epigenetic modifier mutations, tumor suppressor genes, signal pathway mutations, cohesin complex mutations, transcription factor mutations, adhesion complex mutations, mRNA splicing factor mutations, chromosomal abnormalities, ELN score, CD11b/DR and CD11b/CD13 phenotypes, decreased platelet counts, decreased white blood cell counts, etc. and prognosis (e.g., survival time).

  1. Due to the small group of patients and the very different picture of mutations in our patients, it is very difficult to draw conclusions about the impact on prognosis (or survival time). Due to the small patient cohort and the highly heterogeneous mutational profiles, it is difficult to draw any conclusions regarding their impact on prognosis or survival. In the following paper, we focused on initial diagnostics, specifically exploring new correlations between prognostic factors at the time of MDS diagnosis.. We attempted to correlate NGS and FCM images with the R-IPSS/M-IPSS, but we did not obtain any statistically significant observations. The results are available in the proper section. A comment on this topic can be found in the discussion . We are currently continuing to monitor survival/response to azacitidine therapy, as well as the impact of baseline immunophenotype and mutations on survival. We hope to report our results soon.

Reviewer 2 Report (New Reviewer)

Comments and Suggestions for Authors

Comments to the Author(s):
This study attempts to establish a correlation between the diagnostic parameters established by ELN flow cytometry and the mutational status. This data is of the utmost importance for an adequate diagnostic approach in MDS. However, after reviewing the study, some questions and comments arose:
1.    Line 252 mentions that the study included 30 patients diagnosed with high- or intermediate-risk MDS, but Table 1 shows that IPSS-R very low- and low-risk cases were included.
2.    Line 261 mentions Table 3, which is not included in the manuscript. Perhaps you are referring to Supplementary Table 3.
3.    In Table 1, the IPSS-R risk stratification shows N = 27, but the total number of cases is only 26.
4.    The basal characteristics are not present in all cases. The text does not explain the implications of this data omission.
5.    Figure 2 is scattered and could be combined into a single page to improve comprehension.
6.    Line 294 mentions that IPSS-M is correlated with the ELN 2012 score, yet no results are shown in the manuscript. However, some data is included in supplementary table 3. Furthermore, the discussion from lines 336 to 343 does not include this as a relevant finding.
7.    This study aims to improve the diagnostic approach in MDS; however, the cases were not classified using the latest WHO classification or ICC of 2022.

Author Response

This study attempts to establish a correlation between the diagnostic parameters established by ELN flow cytometry and the mutational status. This data is of the utmost importance for an adequate diagnostic approach in MDS. However, after reviewing the study, some questions and comments arose:

We would like to thank you for your exceptionally thorough and detailed review of our work, and for your very important comments that improve the quality of our manuscript.

1.   Line 252 mentions that the study included 30 patients diagnosed with high- or intermediate-risk MDS, but Table 1 shows that IPSS-R very low- and low-risk cases were included.

Response: We agree that there was an inaccuracy in this sentence. The sentence has been corrected.
2.    Line 261 mentions Table 3, which is not included in the manuscript. Perhaps you are referring to Supplementary Table 3

Response: Thank you for pointing this out. The reference has been corrected - we indeed meant Supplementary Table 3. We  corrected this mistake.

3.   In Table 1, the IPSS-R risk stratification shows N = 27, but the total number of cases is only 26.

Response: We made a mistake when rewriting data from the database. We are very sorry about that. Corrected in the revised version of our manuscript.
4.    The basal characteristics are not present in all cases. The text does not explain the implications of this data omission.

Response: We have updated the description in the results section, according to the reviewer’s suggestion.

5.   Figure 2 is scattered and could be combined into a single page to improve comprehension.

Response: Figure 2 presents a lot of scattered data, and  we are concerned that reducing size of these graphs and combining them into a single page may compromise readability. can make the data harder to read. However, we leave the final decision regarding the size and arrangement of these figures to the editor’s expertise and discretion.

6.   Line 294 mentions that IPSS-M is correlated with the ELN 2012 score, yet no results are shown in the manuscript. However, some data is included in supplementary table 3. Furthermore, the discussion from lines 336 to 343 does not include this as a relevant finding.

Response: We attempted to correlate M-IPSS with ELN 2012, but we did not obtain statistically significant results, which we also write about in the discussion. We supplemented the discussion with information about M-IPSS.

  1.    This study aims to improve the diagnostic approach in MDS; however, the cases were not classified using the latest WHO classification or ICC of 2022.

Response:

Most patients were recruited before or during the transition to the new WHO/ICC 2022 classification. Due to potential challenges with reclassification, we chose to use the WHO 2016 classification to ensure consistency. We believe this decision does not significantly impact the main objectives or conclusions of our study.

Round 2

Reviewer 2 Report (New Reviewer)

Comments and Suggestions for Authors

The changes made are sufficient to address the concerns, but the overall content remains comparable to the previous version. The final decision regarding the manuscript will be determined by the editors.

Author Response

Dear Reviewer,
Thank you again for your thorough review of our manuscript and your valuable comments.

Sincerely, 

Maciej Majcherek

This manuscript is a resubmission of an earlier submission. The following is a list of the peer review reports and author responses from that submission.

Round 1

Reviewer 1 Report

Comments and Suggestions for Authors

The manuscript is dedicated to the prognostic value if the somatic mutations to the degree of the MDS. Authors provide a brief summary of that and conclude that mutations are not directly explain the degree. The study is far to short and need further analysis. For instance authors could apply a clustering and classifications tools, also machine learning to test all the measured parameters they could collect including age, gender, clinical history and so on. 

Comments on the Quality of English Language

Some sentences and words must be fixed before publication, like precursors (not prekursors). Header DNA isolation DNA. and alike. 

Author Response

Reviewer 1 comment: The manuscript is dedicated to the prognostic value if the somatic mutations to the degree of the MDS. Authors provide a brief summary of that and conclude that mutations are not directly explain the degree. The study is far to short and need further analysis. For instance authors could apply a clustering and classifications tools, also machine learning to test all the measured parameters they could collect including age, gender, clinical history and so on. 

Response:

Agree.

The study was originally intended to look for a relationship between the occurrence of mutations detected by NGS and immunophenotype. In further steps, we are also planning to look for correlations between the presence of mutations and the course of the disease and the prognostic factor R-IPSS. Of course, due to the small number of patients included, the possibility of drawing conclusions is limited. By publishing these preliminary results, we would like to encourage further research in this area. In the future, we would like to engage modern techniques in data analysis(AI, ML). Due to the reviewer's comments, we added a paragraph in the discussion describing the problems resulting from the small size of the group (page 9).

Reviewer 2 Report

Comments and Suggestions for Authors

The authors correlated molecular abnormalities and cellular immunophenotypes in 30 MDS patients. The study is well designed, the results are clearly summarized, and the methods are comprehensively described. I recommend this study for publication in the International Journal of Molecular Sciences.

Minor comments and questions:

  1. The study focuses on mutations and immunophenotypes, whereas cytogenetic aberrations appear underrepresented. Since the authors used the IPSS‑R, cytogenetic risk is at least partly accounted for.
  2. In Table 1, please specify which WHO classification version you applied. You list subtypes according to WHO 2022 in the table, but on page 2 you cite the WHO 2016 criteria.
  3. When citing R‑IPSS and M‑IPSS on page 2, reference [10] is not the original publication of these scoring systems. If you wish to keep [10], you could for example clarify that reference [10] was chosen because your aza‑eligible patients were classified as MDS/AML.
  4. Have you looked for any correlation between immunophenotypic profiles and IPSS‑M scores? In the discussion (page 9), have ELN 2012 scores ever been correlated with IPSS‑M in the literature?
  5. On page 6, is the designation “high‑risk” or “intermediate‑risk” MDS based on IPSS‑M or on IPSS‑R?
  6. In Table 2, the first four lines are also presented in Table 1. To avoid redundancy, you might consolidate these data into a single table.
  7. On page 8, you note that patients with TP53 mutations had higher ELN 2012 scores. Could TP53 status serve as a surrogate for complex karyotype or for high‑risk prognosis under IPSS‑M or IPSS‑R? Was the TP53 variant allele frequency >20%, thus meeting the “high‑risk” threshold in ELN and IPSS‑M? Were TP53 mutations associated with complex karyotypes or with isolated del(5q) in your study? Which gene was mutated in the sixth patient with tumor suppressor gene mutations?
  8. Does “IDH” refer to IDH1 and IDH2 mutations?

Author Response

The authors correlated molecular abnormalities and cellular immunophenotypes in 30 MDS patients. The study is well designed, the results are clearly summarized, and the methods are comprehensively described. I recommend this study for publication in the International Journal of Molecular Sciences.

Response: First of all, thank you very much for your substantive comments, which will help to increase the value of our publication. We are very pleased to hear that we could interest the reviewer in our research

Minor comments and questions:

  1. The study focuses on mutations and immunophenotypes, whereas cytogenetic aberrations appear underrepresented. Since the authors used the IPSS‑R, cytogenetic risk is at least partly accounted for.

We have added information on the results of cytogenetic testing obtained in our patients to Table 1.

  1. In Table 1, please specify which WHO classification version you applied. You list subtypes according to WHO 2022 in the table, but on page 2 you cite the WHO 2016 criteria.

We have corrected table 1

  1. When citing R‑IPSS and M‑IPSS on page 2, reference [10] is not the original publication of these scoring systems. If you wish to keep [10], you could for example clarify that reference [10] was chosen because your aza‑eligible patients were classified as MDS/AML.

We changed reference [10] to 11 and 12 referring to the original publications.

  1. Have you looked for any correlation between immunophenotypic profiles and IPSS‑M scores? In the discussion (page 9), have ELN 2012 scores ever been correlated with IPSS‑M in the literature?

The results state that the M-IPSS score did not correlate with the ELN 2012 Scale results. So far, we have not found any publications on this subject. Our study group was dominated by patients from the highest risk groups, which may be why we were unable to demonstrate a correlation between ELN 2012 and R-IPSS or M-IPSS.

  1. On page 6, is the designation “high‑risk” or “intermediate‑risk” MDS based on IPSS‑M or on IPSS‑R?

Based on IPSS-R. Text was corrected.

  1. In Table 2, the first four lines are also presented in Table 1. To avoid redundancy, you might consolidate these data into a single table.

Duplicate data has been removed from Table 1.

  1. On page 8, you note that patients with TP53 mutations had higher ELN 2012 scores. Could TP53 status serve as a surrogate for complex karyotype or for high‑risk prognosis under IPSS‑M or IPSS‑R? Was the TP53 variant allele frequency >20%, thus meeting the “high‑risk” threshold in ELN and IPSS‑M? Were TP53 mutations associated with complex karyotypes or with isolated del(5q) in your study? Which gene was mutated in the sixth patient with tumor suppressor gene mutations?  

Due to the small size of the study group, we could not correlate individual mutations with cytogenetic and immunophenotypic changes. Of course, the presence of TP53 mutations is associated with poor prognosis, which is reflected in the M-IPSS classification. In our group, all but one patient with a TP53 mutation had a complex karyotype. This is a very interesting observation. We added it to the results. We entered the specific information on mutation in Table 3.

  1. Does “IDH” refer to IDH1 and IDH2 mutations?

It refers to IDH1. Text was corrected.

Response: First of all, thank you very much for your substantive comments, which will help to increase the value of our publication. We are very pleased to hear that we could interest the reviewer in our research

Reviewer 3 Report

Comments and Suggestions for Authors

This study integrated next-generation sequencing (NGS) and multiparameter flow cytometry (MFC) to analyze the gene mutation profiles and immunophenotypic characteristics of patients with myelodysplastic syndromes (MDS), aiming to explore their potential value in the diagnosis and prognostic assessment of MDS. Overall, this study has certain clinical significance, and the data analysis is relatively comprehensive.

Author Response

This study explored the correlation between gene mutation profiles and immunophenotypic characteristics in patients with myelodysplastic syndromes (MDS), using next-generation sequencing (NGS) and multiparameter flow cytometry (MFC) for integrated analysis, with the aim of evaluating their potential value in MDS diagnosis and prognostic assessment.

Response to reviewer:

Thank you very much for reading our work so thoroughly. We received a lot of valuable comments from the reviewer, which, when supplemented, will certainly increase the value of our work.

To further improve the clarity and completeness of the manuscript, the following suggestions are provided:

  1. It is recommended to specify the inclusion and exclusion criteria for patient enrollment in the Methods section, and clarify whether there is any potential selection bias in the sample.

The inclusion criteria covered all patients diagnosed with MDS qualified for azacitidine treatment. The only exclusion criterion was the patient's lack of consent to participate in the study. We included consecutive patients who gave informed consent. They were mainly patients with high and very high risk according to R-IPSS (total 54%). In the discussion we mention that this is a certain problem when correlating, for example, with the ELN 2012 result.

Following the reviewer's recommendations, we have supplemented the information in the description of the methods.

  1. Please clarify whether informed consent was obtained from the patients or their legal guardians, and provide the corresponding ethics approval number.

Following the reviewer's recommendations, we have supplemented the information in the method description and in the footer.

  1. It is suggested to provide more detailed information on the mutation profiles, such as mutation frequency, mutation types, and distribution of mutation sites, as well as their specific association with immunophenotypic markers, preferably presented in a table.

The data were presented in table 3 in supplement.

  1. It is recommended to further strengthen the mechanistic interpretation of the results, for example, why specific mutations (such as epigenetic regulators or tumor suppressor gene mutations) are associated with higher ELN scores.

Of course we agree with reviewer that presenting the mechanisms by which molecular changes affect phenotype would be very interesting. However, so far we have not come across publications that could explain these phenomena.

  1. Please supplement the discussion on the limitations of this study, such as the small sample size, single-center study design, and lack of an independent validation cohort.

The discussion was supplemented according to the reviewer's recommendations

Response to reviewer:

Thank you very much for reading our work so thoroughly. We received a lot of valuable comments from the reviewer, which, when supplemented, will certainly increase the value of our work.

Round 2

Reviewer 1 Report

Comments and Suggestions for Authors

Authors admit that their findings are based on far too small group of data, therefore preliminary, therefore unreliable  and not worth publishing (the latter is mine). 

Author Response

Authors admit that their findings are based on far too small group of data, therefore preliminary, therefore unreliable  and not worth publishing (the latter is mine). 

Thank you very much for this opinion, we  appreciate the time spent preparing the review.. The group is indeed small, but we still hope that the observations made seem valid, well-documented and worthy of publication to get researchers interested in this topic.

Reviewer 2 Report

Comments and Suggestions for Authors

The authors have answered all of my questions to my full satisfaction. They may considerformatting all gene names in italics. I recommend the manuscript for publication.

Author Response

The authors have answered all of my questions to my full satisfaction. They may considerformatting all gene names in italics. I recommend the manuscript for publication.

Once again, thank you for your comments and contribution to improving the publication. The table has been corrected.